# Pollution and Heritage Metals—Effectiveness of Oddy Testing and Mitigation

**DOI:** 10.3390/ma16247596

**Published:** 2023-12-11

**Authors:** David Thickett, Tayba Batool

**Affiliations:** English Heritage, London SE10 8QX, UK; tayba.batool@english-heritage.org.uk

**Keywords:** pollution, metal, testing, mitigation

## Abstract

Metals are particularly sensitive to some pollutant gases. Many museum showcases and store rooms present atmospheres that can corrode cultural heritage artefacts containing metals. Whilst numerous strategies have been reported to mitigate such situations, avoiding them is preferable. Several approaches to testing materials used in construction, fitting out or dressing are used. The relative merits and drawbacks are discussed. Several parameters of the most widely used, accelerated corrosion ‘Oddy’ test are investigated. The influence of abrasive on subsequent corrosion of lead and copper coupons are assessed. Quantification methods for tested coupons are reviewed. The influence of test duration and temperature are assessed through comparison with real-life, long-term experience of material behaviour. Direct contact tests with touching the test material are investigated. Several materials present in artefacts are known to potentially cause corrosion when enclosed with metals in other artefacts. A number of situations are investigated with pollution and RH and some corrosion rate measurements. Ways to isolate artefacts or mitigate are explored and a decision support model is further developed.

## 1. Introduction and Aims

Metals are particularly sensitive to some pollutant gases. Many museum showcases and store rooms present atmospheres that can corrode cultural heritage artefacts containing metals. Most frequently, the gases causing corrosion originate from off-gassing of materials present. Whilst numerous strategies have been reported to mitigate such situations, avoiding them is preferable. Several approaches to testing materials used in construction, fitting out or dressing, are used. The first is the accelerated corrosion test described by Oddy in 1973 [1]. It involves enclosing 2 g of test material with silver, copper and lead coupons at 60 °C and near 100% relative humidity, RH for twenty-eight days. After the aging period, the amount of corrosion on the coupons is assessed visually. Various schemes are used, a common one is suitable for permanent use, suitable for temporary use and unsuitable for use. The test (and others developed) is best considered as indicating risk from pollution. Whether pollution leads to corrosion in the actual situation depends on a number of factors. The loading of the material in the enclosure or the storeroom; the temperature and relative humidity; the RH and light levels for some materials; the air exchange rate (AER) of the showcase or the room. The relationship between these factors and the concentration developed are described by the Weschler and Shields equation [2]. Whether this concentration causes corrosion depends on the metals present, their composition, and the RH. The amount of risk tolerated is probably institution dependent and should consider the location’s background corrosion rate. In a museum with extensive heating, ventilation and air conditioning and chemical filtration the corrosion rate are low, and that in enclosures should be lower, whilst less well-funded institutions with natural ventilation and no chemical filtration, on the whole, have higher room corrosion rates. A very large range of corrosion rates in heritage institutions has been reported [3]. In the latter situation, a strict sifting of materials causing very slight amounts of corrosion in Oddy tests could be considered excessive, as the artefacts corrode at a certain rate anyway. Alternatively, perhaps protection by showcases is even more important, but different approaches to assessing an acceptable level of corrosion on the coupons exist.

The test method has been continuously improved [4,5,6]. One of the strengths of the test is it does not require expensive analytical equipment and could in theory be undertaken by any reasonably equipped conservation facility. The empirical nature of the test is also beneficial; if a material causes corrosion, it is observed, and the identity of the problematic gases does not need to be known. Its main drawback is the time required for the test (28 days) and its perceived variability in results. Many groups have modified the parameters of the test to suit their situation. For any test, this introduces an enormous amount of variability in response. Indeed, research has indicated that test results are sensitive to small variations in method [7]. With a well-defined method, the two parameters that have most significant effects are the metal coupon preparation method and the assessment of the tested metal coupons. Korenberg introduced abrasive cloths into the method in 2017, replacing the previous glass bristle brush, GBB abrasion [6]. This has undoubted advantages; it appears to generate less lead dust and no sharp glass bristle fragments. The two methods were investigated by asking a group of experienced practitioners to prepare lead and copper coupons with both methods and then testing the surface roughness and relative corrosivity.

There have been attempts to improve the standardisation of the test by using set equipment and pre-prepared metals [8]. Unfortunately, this is very likely to increase the cost per test, limiting its accessibility and reducing the number of materials tested. It may be that two types of tests are required: a simple, inexpensive version that can be widely used and a more controlled version, possibly using analytical instrumentation that can be used in larger institutions.

Several methods have been used to improve quantification of the corroded coupons. Reference photographs and descriptions were developed [9] and have been extended and refined [10]. The several analytical methods developed are summarised in Table 1 from the literature and observed practice. Some techniques have been used for coupons used to assess atmospheres and not from Oddy tests, but it is assumed that the challenges are the same. Many analytical techniques suffer from a mismatch between the analytical area and the rectangular coupons used in the test. As the corrosion can be uneven, ideally, the whole coupon surface needs to be analysed.

All the techniques listed have proved to be able to quantify corrosion, at least on some coupon types. The corrosion layers on silver are often very thin and require sensitive techniques to detect them. Clearly, such techniques are only available in well-equipped laboratories. Chemical stripping is likely to be possible in many conservation facilities, although the solutions used are often toxic. To use it quantitively, for Oddy test coupons, a high accuracy balance is required, generally beyond those present in conservation workshops. Additionally, the solution recommended for lead in ISO 11844 [29], ammonium acetate, has a significant vapour pressure of acetic acid. Depending on extraction used, room volume, and leakage, this could present a risk to lead artefacts undergoing conservation in the same space. Electrochemical stripping is more accessible than many of the other techniques and frequently used in metal conservation. Cano et al. have suggested the use of coulometric stripping as per ISO 11844 [21,29]. However, potentiodynamic stripping requires less capable (hence, less expensive and more accessible) potentiostats and does not necessarily require nitrogen purging, which most conservation facilities do not have access to. Lead is an issue with either electrochemical technique (not covered by ISO 11844), with some corrosion products having general solubility in the stripping solutions and multiple peaks often being generated even from simple corrosion products. Experiments with silver and copper coupons and comparison with visual assessment have been caried out.

Several researchers have investigated analysing the gases emitted from a material and then determining whether they cause corrosion. Many showcase and store environments contain hundreds of gases. So far, 33 gases have been reported to cause problems with different heritage materials, and this number has increased rapidly over the past two decades [30]. It is almost certain there are other gases, yet unidentified, that are problematic, and to work fully, this approach requires a perfect knowledge of what to avoid. An unfortunate example is Piperodinol gases emitted from two adhesives in showcases. It appears they were detected in emission tests, but as unreported as damaging, the materials were passed and caused significant issues in hundreds of showcases [31,32]. One of the two materials had already failed Oddy tests with copper. Some researchers have suggested using emission tests to screen materials, with those not emitting known problem chemicals then undergoing longer Oddy tests.

Most of the emission tests [33,34,35,36,37,38,39,40] measure room temperature and middle RH emissions. This could be considered a risk, as many of the species identified as problematic are deterioration products of the materials tested and may not be present in the freshly produced material.

Additionally, increasing sensitivity of detection methods to speed up the testing is a common theme [22]. Although mass spectrometry is extremely sensitive, most configurations cannot detect formic acid, and several researchers still use ion chromatography for acetic acid as well [34,41]. Even for very sensitive techniques, there is strong evidence that some materials do not emit the damaging gases until late on, even in accelerated aging [42]. If a gas is not present, it cannot be detected or cause corrosion that can be detected, no matter how sensitive the technique. More sensitive analytical techniques such as SEM have been used on so-called SMART tests run at lower temperatures and RHs [22]. The test runs at 30 °C and 60% RH for 14 days, but some have been reported at 22 °C. Many situations in historic interiors have RH in excess of 60%, and this affects emission from materials [42]. One group of materials (Vinyls) surprisingly passed these tests. Curran et al. reported acetic acid emission from these materials at levels that would be expected to corrode lead [43], and over twenty vinyls have been rated as temporary or worse with Oddy tests. This was investigated with an Oddy test for one material and a series of lower-temperature tests, with material supplied by Liverpool Museums. This was rated as non-corrosive towards lead in a SMART test. The material was enclosed with a cleaned lead coupon at 30 and 20 °C and observed, and corrosion was analysed when formed. The emission of acetic acid from a further series of vinyl samples passing SMART tests with lead was measured.

A series of investigations into corrosion on display or in storage have examined whether Oddy tested materials are present in the enclosure, its AER, analysis of some common gases, acetic and formic acid, hydrogen sulfide and ammonia. The corrosion products were also analysed along with the timeframe over which corrosion was observed. The sometimes very long-term exposures were useful to investigate the success or otherwise of the Oddy tests used.

Even with comprehensive testing of showcase and display materials, certain object materials can have negative effects on other objects in some enclosures. This appears only intermittently in the literature and collections are put at risk by their custodians being unaware of the issue in all instances. A long list of reactions has been observed, and they are collated.

Similar investigations into situations with damage observed were undertaken. Whilst the responses of different metals to pollution are quite well documented, the likely amounts of pollution emitted from objects in different situations are largely unresearched.

A decision support model was developed in the MEMORI (Measurement, effect assessment and mitigation of pollutant impact on moveable cultural heritage) project [44]. It covers both assessment and mitigation options, and a series of examples in these situations have been explored. The model was improved through this work, and an updated version is available [45].

A very large number of interventions are possible. In situ effectiveness depends on many factors. It is critical that after any mitigation, measurements are repeated to ensure it has had sufficient effect. A decision support model was developed as a downloadable, interactive guide using commercial presentation software, which leads users through the options [44,45]. There is a series of spreadsheets to allow calculations to compare the likely effectiveness of options. An overall mitigation decision support model spreadsheet allows comparison between options from answers to twenty-one simple questions and, depending on outcomes, running some of the aforementioned spreadsheets with the user’s environmental and enclosure data. The outcomes are ranked in terms of approximate likelihood of success, rough costs, and indicative carbon footprints. More accurate carbon footprints can be developed from modified versions of the spreadsheets in the tools [42]. This approach, using widely available software, was adopted to ensure longer-term accessibility for the data, as web sites and apps need extensive updating.

The measures are presented as a hierarchy, developed in the MEMORI project [44] from Tetreault’s thoughts [46]. The higher-level strategies are more robust and less likely to fail. The measures are shown in Table 2.

Removal is often impossible, especially if both materials occur in a single object. But within a showcase, there is sometimes the possibility to select other non-reactive suitable objects, depending on the interpretation message and collection. Similarly, blocking emissions is only infrequently possible, generally within enclosures such as glazed frames and Daguerreotype packets. Dilution through increasing air exchange rate can work in some, but not all situations. A simple tool to aid selecting and sizing the required air exchange rate has been developed [44]. The tool is based on the well-validated Weschler and Shields equation [2]. The equation requires material loading, its emission rate and deposition velocity, showcase volume and air exchange rate. The volume is straightforward to determine, air exchange rates can be measured, and loading can usually be estimated. Emission rate data are limited but growing, and a selection of deposition velocities are tabulated in the tool.

Passive incorporation of sorbents can work with copper, iron and lead but is unlikely to be successful with silver. The use of pumps can overcome this in some situations with careful design of airflows. Lowering temperature can control the off-gassing from particularly problematic materials such as some plastics for storage situations.

The aim of this work was to investigate the effect of parameters on accessible corrosion tests and suggest improvements. A large number of instances of corrosion on display were investigated to test the long-term performance of the Oddy test and identify the relative occurrence of corrosion from off-gassing from artefacts that clearly cannot be tested in such a way. Information about situations such as object type, corrosion type, species concentration, climate parameters and showcase AER was collected to inform those responsible for metals collections about likely corrosive situations. Additionally, the utility of tools developed during the MEMORI project and extended recently was assessed using the data acquired.

## 2. Materials and Methods

### 2.1. Coupon Cleaning Tests

Five coupons of lead and copper were cleaned with glass bristle brushes and micromesh by experienced practitioners from four institutions. Three institutions had one practitioner each, one had two. The coupons were transported in a Stewart (London, UK) Gastronorm box with dried silica gel (Gastronrom is a European standard for box sizes, but Stewart use the name for a range of polypropylene boxes).

Coupons from three institutions were analysed with a TraceiT (ADD) profilometer. Three areas of each coupon were analysed by tracing 1500 lines, each 5 mm in length.

The coupons were weighed (balance) and suspended from a 0.05 mm nylon monofilament around the rim of Bernadin (Ottawa, ON, Canada) Mason 250 mL jars. The copper coupons were exposed to 50,000 µg/m^3^ of acetic acid generated above a water solution at 30 °C. The lead coupons were exposed to 1000 µg/m^3^ of acetic acid. The tests were checked daily. When sufficient corrosion developed, the coupons were removed and stored above dried silica gel for seven days. The coupons were weighed again, and corrosion products were determined with external reflection and diamond attenuated total reflectance (ATR) on a Bruker (Coventry, UK) Alpha FTIR spectrometer.

### 2.2. Oddy Tests

Oddy tests were carried out according to the methods described in 1995 and 2003 [5], with a micromesh 1200 fabric used to clean coupons after 2017. Direct contact tests were carried out by modifying the most recent method. Smaller coupons (20 by 10 mm) were cleaned as usual. They were wrapped in the fabric being tested or placed in direct contact with powder-coated metal pieces. Close contact was assured by placing the samples in a polyethylene terephthalate fitting. The fitting is required for test tube-based tests but can be omitted for flask- or conical flask-based tests. Fabric and powder-coated samples were assessed as these are most likely to be in direct contact with artefacts.

For materials from showcases investigated for corrosion where 2 g was not available, a miniaturised test was developed. This included a small (5 mm by 5 mm) sample of the metal observed to corrode, cleaned as before, in a 5 mL glass vial (Fisherbrand 13 mm crimp neck vial) with a crimped aluminium lid, a Duran tube with 0.2 mL of water, and 0.2 g of the text material. The Duran tube was placed between the sample and the metal coupon. The other test details were identical.

The sensitivity of the Oddy test with acetic and formic acid was determined by introducing 0.5 mL of solution into a test with lead and running as normal. Solutions of 0.05, 0.1, 0.2, 0.4, 0.6, 0.8, and 1 µg/L were used.

### 2.3. Coupon Assessment with Potentiodynamic Stripping

Silver and copper coupons from previous tests were analysed with a Palmsens (Houten, The Netherlands) 3 potentiostat, a silver/silver chloride reference electrode and a platinum counter electrode. Silver was placed in 1 M sodium nitrate electrolyte, attached with a crocodile clip above the solution using the portion of the coupon that was in the stopper. Copper was similarly placed in a 1 M sodium carbonate,1 M sodium hydrogen carbonate solution. The potential was swept from 0 to −1.4 V (vs. standard hydrogen electrode).

### 2.4. Lower-Temperature Corrosion Tests

Vinyl supplied by Liverpool Museums was cut into 2 g pieces, equilibrated to 50% RH for 3 weeks and placed in Oddy-type tests with lead (same containers, but at lower temperature and with no added water). Tests were run at 30 °C or 20 °C in a Memmert (Schwabach, Germany) HPP260 eco-environmental chamber.

### 2.5. Emission Tests

Tests were undertaken in a 250 L stainless steel chamber controlled to 23 + 1 °C and 50 + 2% RH in the Memmert environmental chamber. They were performed in accordance with BS EN ISO 16000, Part 9 [33]. The air flow rate was set to 10 L per hour using a Cassela pump. This is a low value that approximates to an air exchange rate of 1 per day. The vinyl samples (40 by 60 cm) were preconditioned at 23 °C and 50% RH for 72 h before the tests. Air was sampled through 0.1 M sodium hydroxide (120 L at 2 L/min, same type of pump) after 1, 4, 24 and 72 h. The sodium hydroxide solution was analysed with a Thermo (Crawley, UK) ICS1100 ion chromatograph with an AS12A column and a 18 mM sodium hydrogen carbonate and sodium carbonate solution. The results were used with the MEMORI air exchange rate (AER) tool to calculate the AER and surface loading combinations that would result in an acetic acid concentration exceeding 400 µg/m^3^ which poses a risk to lead at 50% RH.

### 2.6. Corrosion Analyses

Instances of observed corrosion were analysed with attenuated total reflection Fourier transform Infra-red spectroscopy, ATR FTIR (Nicolet (now Thermo) Avatar 360 with Inspect IR microscope with silicon ATR, or Bruker Alpha with diamond ATR), X-ray fluorescence (Bruker (Blue Scientific, Cambridge, UK) Tracer III/IV), X-ray diffraction, XRD (Phillips (Cambridge, UK) 1830/1840) and scanning electron microscopy with an energy dispersive X-ray analyser, SEM-EDX (Jeol (Wlewyn Garded City, UK) 840).

Previous Oddy test results were consulted, and any unrecognised materials were tested if samples could be taken. The Oddy test requires 2 g of material, which can be problematic for components of showcases, and a miniaturised test using 0.2 g was used as described previously.

Showcase or storeroom air exchange rates were measured using carbon dioxide tracer gas decay [47]. The air exchange rate of Daguerreotypes was measured using oxygen ingress. The Daguerreotype was flushed with nitrogen for eight hours and then placed in lab air. The oxygen concentration inside the Daguerreotype was continuously measured with a Presens (Regensberg, Germany) Fibrox 4 oxygen system. The very small thickness of the SP-PSt3-NAU-D7-YOP spot allowed analysis inside the shallow Daguerreotype.

Acetic and formic acids and ammonia were analysed with Palmes diffusion tubes (purchased from Gradko, Winchester, UK), exposed for four weeks. The acid diffusion tubes were based on potassium hydroxide sorbent and analysis with a Dionex (no longer trading) 600 ion chromatograph with an AS12A column and 18 mM sodium carbonate/sodium bicarbonate eluent [48]. The tubes were exposed in quadruplicate. The ammonia tubes used sulfuric acid eluent. A Dionex 600 ion chromatograph was used with a CS12 column and 18 mM methane sulfonic acid eluent. In most instances, hydrogen sulfide was analysed using Gradko (Winchester, UK) Palmes diffusion tubes. One set of measurements was undertaken on archaeological ceramics and iron in Tedlar bags at 25 and 35 °C, extracting a gas sample and analysing it with Perkin Elmer (Beaconsfield, UK) 850 gas chromatograph with a 1.8 m 0.3 mm diameter Teflon column with Carbopac B/1.5% XE60/1% H_3_PO_4_ and nitrogen carrier gas. This analysis quantified hydrogen sulfide (detection limit 0.01 ng/m^3^), carbonyl sulfide (detection limit 0.05 ng/m^3^) and carbon disulfide (detection limit 0.02 ng/m^3^).

### 2.7. Mitigation of Corrosive Environments

Mitigation of some of the instances investigated with increasing air exchange rate was investigated with the MEMORI AER-ventilation modelling spreadsheet [44,45]. The approaches towards pollution mitigation were investigated with the Mitigation decision support model spreadsheet for a subset of data with acetic or formic acid as the cause of corrosion and metals included in the MEMORI decision support model [44]. For one instance, a fuller carbon footprint was developed, considering the embedded carbon and the use over the showcase’s lifetime (set at 10 years) using the method described in [42].

## 3. Results and Interpretation

### 3.1. Cleaning Tests

The arithmetic mean of surface roughness, Ra, was calculated for each area analysed. The results for copper are shown in Figure 1 and Figure 2; those for lead are shown in Figure 3 and Figure 4.

The roughness in the lateral direction was generally lesser than in the longitudinal direction, which is the last direction the coupons were cleaned in. HCP had much lower roughness and less variation with both techniques. MOL is higher and NHM is higher still with GBB, but lower with a micromesh. The micromesh-cleaned coupons showed a greater surface depth of scratches than those cleaned with glass bristle brushes for all three preparators.

Again, the roughness in the lateral direction was generally lesser than that in the longitudinal direction. The micromesh-cleaned coupons showed a greater surface depth of scratches than those cleaned with GBB for both HCP and MOL. They seemed comparable for NHM. HCP had much tighter distribution for GBB-prepared coupons, but this widened when a microfibre cloth was used. The variation was quite similar for the two methods of preparation.

The mass gains recorded with the lead and copper coupons cleaned with GBB and abrasive cloth are shown in Figure 5 and Figure 6.

The coupons showed between 31 and 77% distribution in corrosion for a single preparator ((max value − min value)/max value), with EH and HCP showing lower values and NHM showing the highest. The micromesh-cleaned coupons showed a greater degree of corrosion than those cleaned with GBB, except for NHM. The variation was mainly greater for the micromesh, certainly for HCP, MOL, EH1 and EH2. Lead formate was identified on all coupons.

The coupons showed between 31 and 45% distribution in corrosion for a single preparator ((max value − min value)/max value). Again, the micromesh coupons generally showed a greater degree of corrosion than those cleaned with GBB, especially at HRP. At NHM, they actually showed a lower degree of corrosion. The distributions for the lower corroded coupons HRP, EH1 and EH2 where much tighter (lower range) for GBB than for micromesh, with ranges 1.51, 2.15 and 1.76, cf. 2.89, 2.36 and 2.37. MOL and NHM appeared to have similar distributions for the two techniques. Cuprite and copper formate dihydrate were identified on all coupons.

There was no fully consistent trend in the results considering all participants. It appears the preparation phase varies depending on the person undertaking the test. However, some preparators had much more consistent results, so it is likely that training could probably reduce the variation.

### 3.2. Oddy Tests

The results of the direct contact tests are shown in Table 3.

It is clear that there are a number of materials that pose a risk in direct contact with metal objects that cannot be identified with conventional accelerated corrosion tests. It appears there are non-volatile compounds that can accelerate corrosion in direct contact.

Acetic acid solution causes corrosion visible to the naked eye at 0.8 µg/L, but not 1.0 µg/L; formic acid does the same at 0.4 µg/L, but not at 0.6 µg/L. Acetic acid results correlate well with those reported by Stephens [49]. They are slightly lower, but that work only had divisions of 0.1 and 1 µL/L.

### 3.3. Coupon Assessment with Potentio-Dynamic Stripping

The results for potentio-dynamic stripping of silver coupons are shown in Figure 7, and those of copper are shown in Figure 8.

Generally, the amount of corrosion increased between the visual assessment categories; however, the two metals had significant overlap regions across the permanent/temporary border. For silver, this was also the case across the temporary/unsuitable border, whilst for copper there was a clear boundary. There was a disagreement between the amount of corrosion measured and the visual assessment. In these instances, this was due to localised corrosion, a few spots, or a band of corrosion at or near the edge of coupons. The small amount of corrosion present provided a low value as a result when averaged over the whole coupon area. Cano and Diaz discussed the mismatch that occurs between visual observation and the amount of corrosion in their experiments with four materials [21]. It should be noted that most problematic instances of corrosion are initially identified by visual assessment, and perhaps this should be weighted more heavily A combined method is likely to be required with some visual assessment of how even the distributed or localised corrosion is, followed by analysis. Different ranges would be needed for localised and evenly distributed corrosion on coupons.

### 3.4. Lower-Temperature Tests

The vinyl sample supplied by Liverpool Museums failed duplicate Oddy tests with lead (and passed with silver and copper). White corrosion was visible through the glass vessel after 34 days at 30 °C and after 151 days at 20 °C. The corrosion was identified as basic lead carbonate with XRD and FTIR. No other compounds were identified, but detection limits in mixtures are often estimated at around 20% with both techniques. This is very dependent on the compounds present.

### 3.5. Emission Tests

The emission results from the other vinyl samples are listed in Table 4.

All samples tested emitted acetic acid despite passing the SMART tests with lead. The sample provided by Liverpool Museums had the highest concentration, and this was over the low-risk (observed corrosion in under two years) threshold [44]. Considering Tetreault’s reported lead corrosion rates, the behaviour is consistent with visible corrosion appearing, but not for some period [23]. The ventilation rate calculations show that all the tested vinyls generate acetic acid concentrations that pose a risk to lead under some circumstances. Many recent showcases have been built to an AER specification of 0.10 per day. The vinyls are used to hold graphics in showcases. The tests in Table 4 were conducted for a temporary exhibition, lasting 6 months, and two were actually used. Lead coupons added to the showcase to monitor that atmosphere showed no corrosion after this period at 50% RH. The test appears suitable for short-term exhibitions, but risks corrosion for lead with longer-term display or storage.

### 3.6. Observed Examples of Corrosion In Situ

Over 730 instances of corrosion were investigated. In just over 600 instances, a material was identified as the likely source of pollution. In stores, the very large number of objects and materials present often made attribution impossible; most of the 130 unattributed instances are explained by this. No instances of corrosion were attributed to a material that passed the Oddy test for that particular metal. There were a number of instances where materials failed Oddy tests and were still used. There were also instances were a material passed with some metals and the documentation stated it could be used in that situation, provided certain metals were not introduced in the future. It was exactly these later introduced metals that corroded. It is clear that robust systems need to be in place to ensure such information is retained and acted on. Organisations tend to take two approaches; a material must pass with everything or only the metals present on the set-up must pass. The latter is often driven by time pressure, particularly for exhibitions. Figure 9 shows the number of instances investigated (within a single showcase) and Figure 10 demonstrates the number of objects observed to be corroded in those instances.

The instances of object emissions causing corrosion on other objects are shown in Table 5.

The AER, pollutant concentrations and corrosion products were included. The number of objects observed to be affected is also tabulated for each location. Numbers in brackets are the author’s estimate of other reported instances. This is very difficult as much data are in grey literature or unrecorded, for example, in presentations. Also, the number of objects affected is only very rarely reported in the literature.

Table 6 lists those instances when it is thought the gases originated from the object with the metal corroding.

The sulphur gas analyses in the Tedlar bags enclosing the iron and ceramic object from the top row of Table 5 are shown in Table 7.

There was some increased concentration of both hydrogen sulphide and carbonyl sulphide at 25 °C and carbon disulphide from the ceramic. There was a very strong increase at 35 °C. The copper objects were on display with no observed black spot corrosion for over 20 years. Extensive corrosion was observed after an exceptionally hot summer when the case temperature reached 35 °C. The results of this experiment correspond with those observations.

### 3.7. Mitigation of Corrosive Environments

For the Japanese lacquer boxes in the top showcase shown, ventilation was considered as a mitigation. The air exchange rate was 0.87/day, the acetic acid concentration was measured at 2564 µg/m^3^. The showcase volume was 1.24 m^3^ and the emissive surface of boxes was estimated at 0.87 m^2^. Figure 11 shows the output from the ventilation tool using a surface deposition velocity of 0.15.

As can be seen, acetic acid and AER measurements fix a point on the graph. For showcase parameters, 0.87/day is slightly over 74% of the maximum concentration (when AER = 0, left hand side of Figure 8). This is equivalent to 2564 µg/m^3^. Increasing ventilation to an AER of 4.00 is expected to reduce acetic acid concentration to 1334. Consulting the MEMORI table for lead at 50% RH, this is well above the 400 µg/m^3^ threshold for low risk of corrosion. Further increase to above 19.40 drops this to below that 400 figure. This figure is extremely unlikely using passive ventilation, and a small pump was fitted with air being drawn through a dust filter. The pump flow rate was set at 0.5 L/min to exceed the desired AER. Diffusion tube measurements showed that acetic acid concentration reduced 321 µg/m^3^. Small pumps are quite reliable, and some used for computers are guaranteed to function for several thousand hours. If mitigation relies on such units, a procedure is needed to identify when the pump or power does fail. In this instance, the exhibition is only scheduled for nine months, and the risk of failure is thought to be vanishingly small.

For lead corroded by a painted sign, the showcase volume was 0.82 m^3^, the sign area was 0.31 m^2^ and the AER was 0.08/day, which generated a concentration of formic acid of 85 µg/m^3^. This is in the medium-risk category on the MEMORI lead 50% chart. Figure 12 shows the output from the calculation for those showcase parameters.

Increasing the AER to 2.6/day is expected to reduce formic acid concentration to below 50 µg/m^3^, the low-risk concentration. This can be achieved with passive ventilation.

The headline mitigation decision support tool results are shown in Table 8. As these are obtained from objects as sources of pollution, in these instances, inside showcases, no data for avoid (it is assumed the objects need to be in the showcase together for curatorial/presentation reasons) or block are included.

The tool generated an indication of the different approaches that are most likely to succeed.

The support tool generates significant extra data about costs and indications about sustainability. Carbon footprints were extended beyond the indications in the tool. Results for the instance described in third row in Table 6 are shown in Table 9.

The ventilation tool indicated that an increased AER of 14.5 is needed to reduce the acetic acid concentration to below 400 µg/m^3^ at 50% RH. This was not considered feasible with passive ventilation. Calculations with silica gel indicated 8 kg can keep the RH below 30% for 12 months, and that 1039 µg/m^3^ of acetic acid is low-risk at this RH. Similarly, a Munters MG50 dehumidifier was shown with the dehumidifier tool to have sufficient capacity to keep the showcase below 30% in that room environment using 34 kWh of electricity per year. The dehumidifier decreases acetic acid concentration, but since it runs intermittently, the reduction is difficult to calculate. A Dynamax 5 pump has a sufficient flow rate to generate the required AER. The showcase is a permanent installation; the likely success was rated at 7% due to the difficulty in recognising a pump failure. The estimated electricity was 32 kWh per year. The case only had limited space for activated charcoal cloth, so the success likelihood was rated at only 20%. A Dynamax 5 filter pump was assessed at a 100% success possibility due to the incorporated warning light for pump failure or filter exhaustion. The pump uses 42 kWh per year. The carbon footprint is calculated from the greenhouse gas equivalent of the electricity supply in the UK, the embedded carbon in the unit and the embedded carbon of any filters multiplied by the observed replacement rate [42].

## 4. Conclusions

The effect of a number of variables related to Oddy and other corrosion tests were clarified and recommendations were made.

The variability in preparation of lead and copper coupons was assessed with both profilometry and controlled corrosion experiments. Some preparators show much lower variability, indicating that training could probably reduce this to a lower level.

The detection limits of formic acid and acetic acid were found to equate to very low solution concentrations of these species, indicating the very high sensitivity of the Oddy tests with lead to these common pollutants.

Initial experiments with potentio-dynamic stripping indicated that it has the potential for quantifying silver and copper Oddy coupons. Further work is required to develop suitable ranges, and two sets at least are required to accommodate localised corrosion phenomena.

One vinyl material tested caused corrosion in under 6 months at room temperature, and the other five vinyls, assessed with emission testing, did emit acetic acid. This caused corrosion in some instances. This emphasises the need for some form of acceleration in testing for long-term display and storage.

The large number of investigated instances of corrosion on display provided very strong evidence of the positive effect of Oddy testing. No instances were observed of materials passing the test and going on to cause corrosion. A relatively small number of instances were attributed to off-gassing from objects and the situations described in terms of corrosion type, gas concentrations and air exchange rates.

When other cultural heritage objects were present that posed a risk of off-gassing, the online ventilation tool developed was shown to predict when this approach to lowering concentration is effective and allow an AER target for assessing the approach. The online decision support model ranked different approaches in terms of likely effectiveness, cost and rough carbon footprint. The potential to undertake, further, more directed carbon footprint calculations was also demonstrated.

## Figures and Tables

**Figure 1 materials-16-07596-f001:**
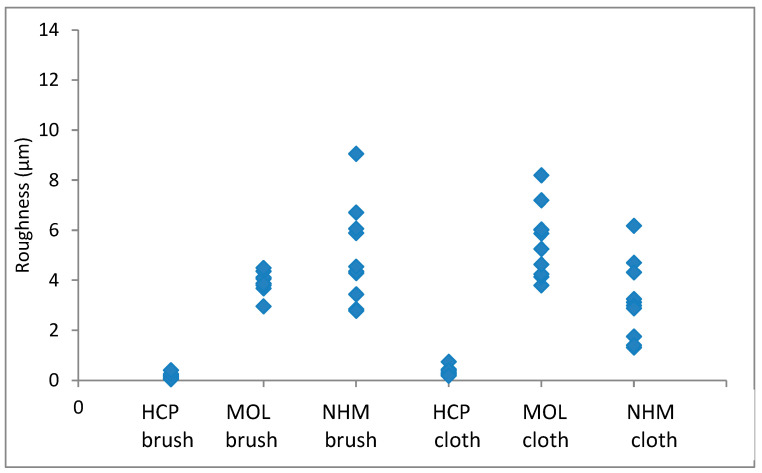
Arithmetic means for surface roughness of copper in directions down the coupon.

**Figure 2 materials-16-07596-f002:**
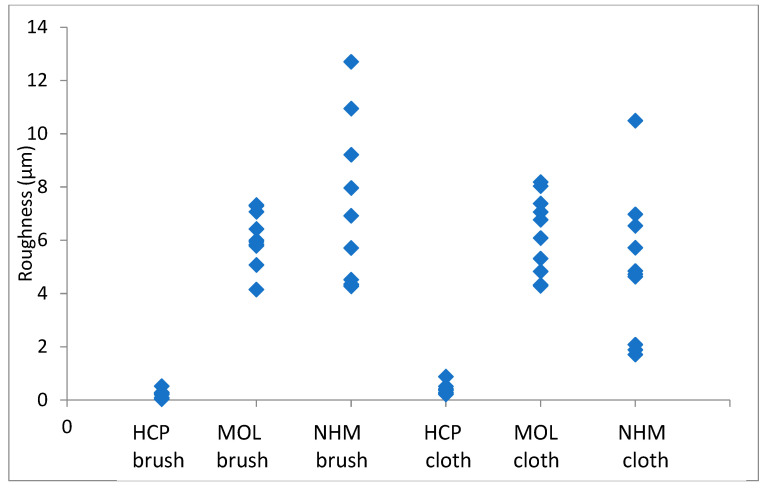
Arithmetic means for surface roughness of copper in directions across the coupon.

**Figure 3 materials-16-07596-f003:**
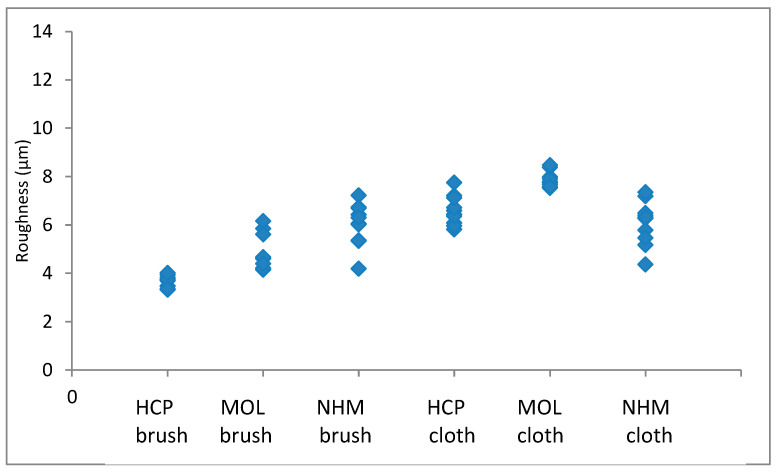
Arithmetic means for surface roughness of lead in directions down the coupon.

**Figure 4 materials-16-07596-f004:**
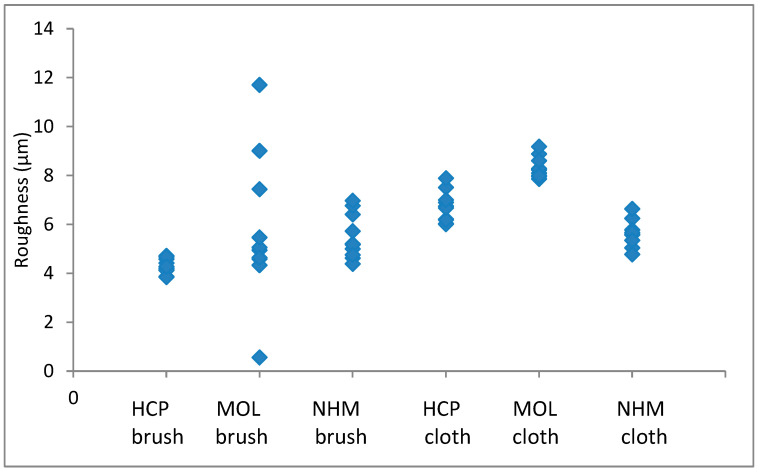
Arithmetic means for surface roughness of lead in directions across the coupon.

**Figure 5 materials-16-07596-f005:**
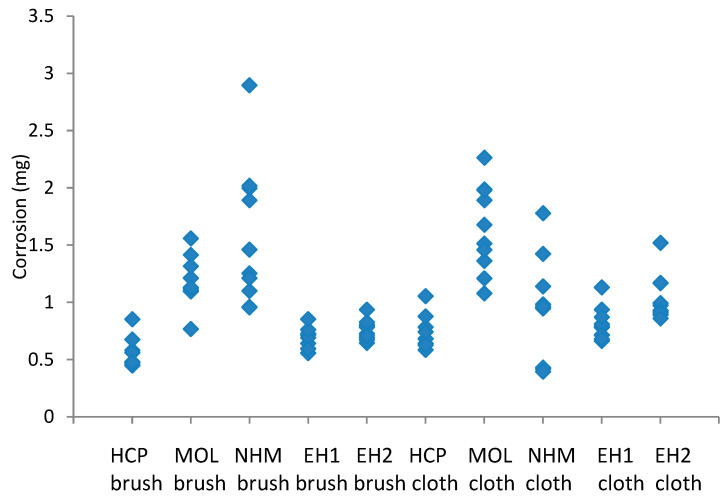
Corrosion formed on lead coupons exposed to 1000 µg/m^3^ of acetic acid, cleaned with glass bristle brush and abrasive cloth at different locations.

**Figure 6 materials-16-07596-f006:**
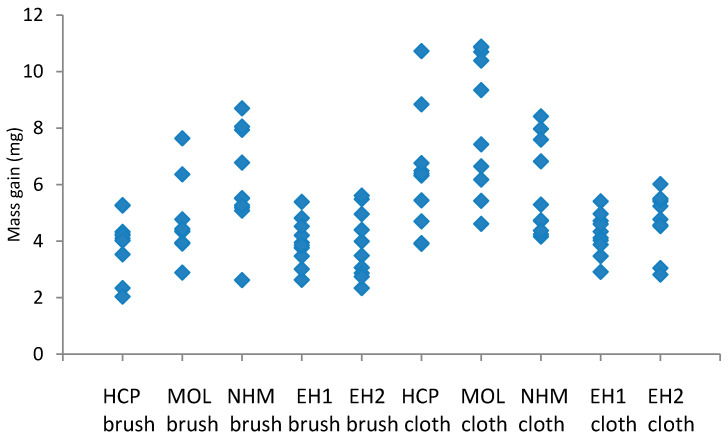
Corrosion formed on copper coupons exposed to 10,000 µg/m^3^ of formic acid, cleaned with glass bristle brush and abrasive cloth at different locations.

**Figure 7 materials-16-07596-f007:**
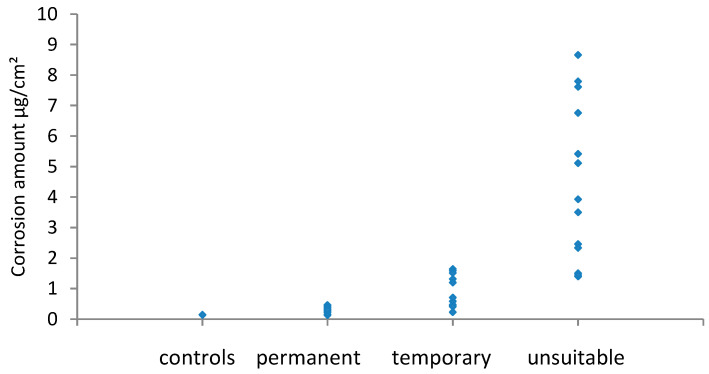
Silver stripping results and visual ratings.

**Figure 8 materials-16-07596-f008:**
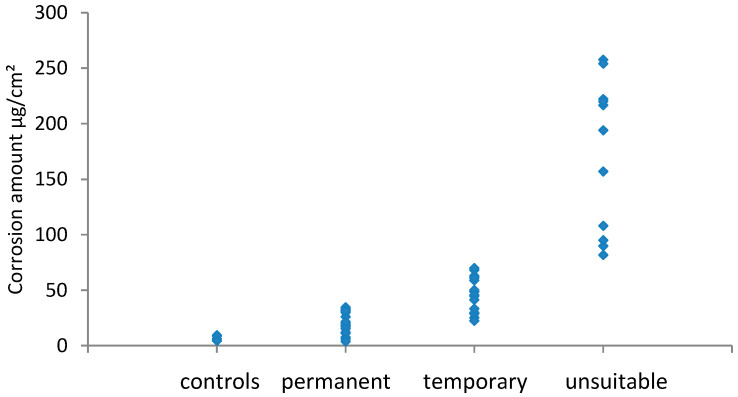
Copper stripping results and visual ratings.

**Figure 9 materials-16-07596-f009:**
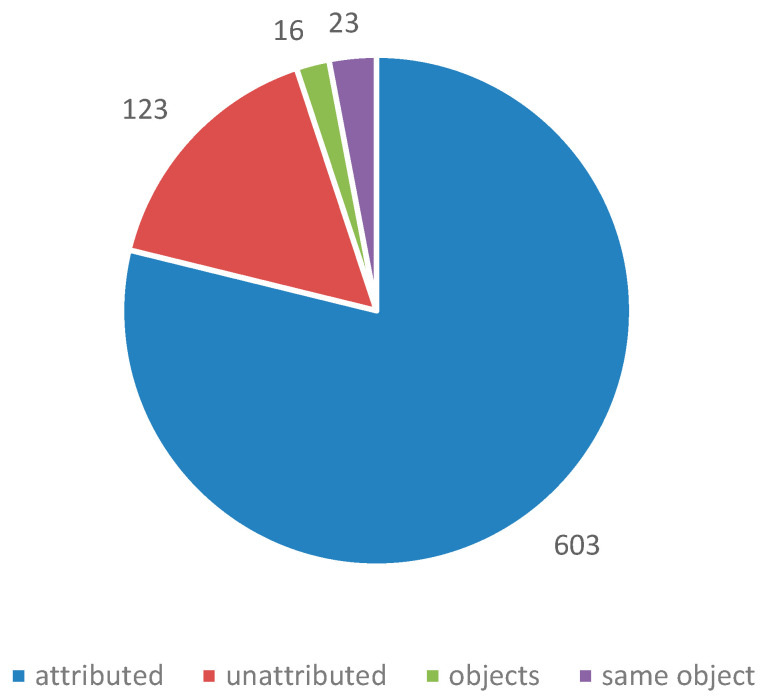
Number of instances of observed corrosion.

**Figure 10 materials-16-07596-f010:**
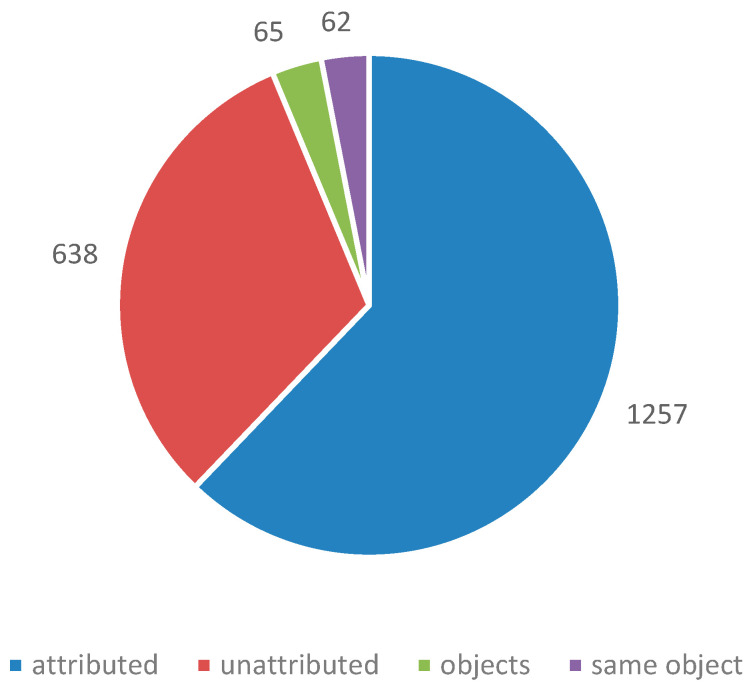
Number of metal objects affected by corrosion.

**Figure 11 materials-16-07596-f011:**
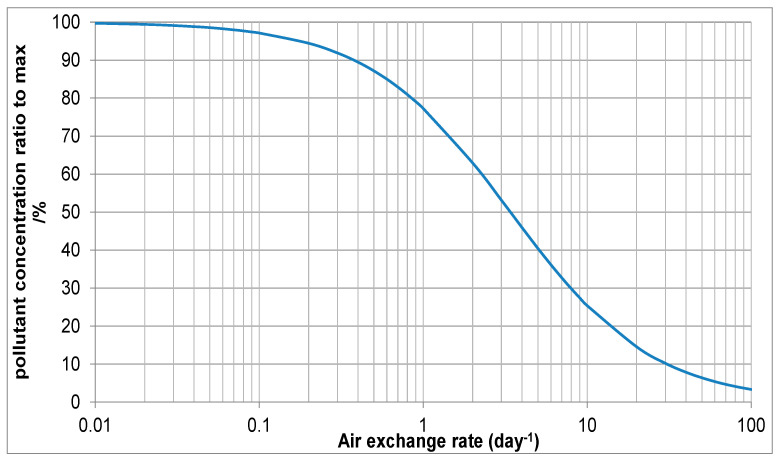
Effect of air exchange rate on acetic acid concentration in the 0.87 AER showcase with Japanese lacquer boxes.

**Figure 12 materials-16-07596-f012:**
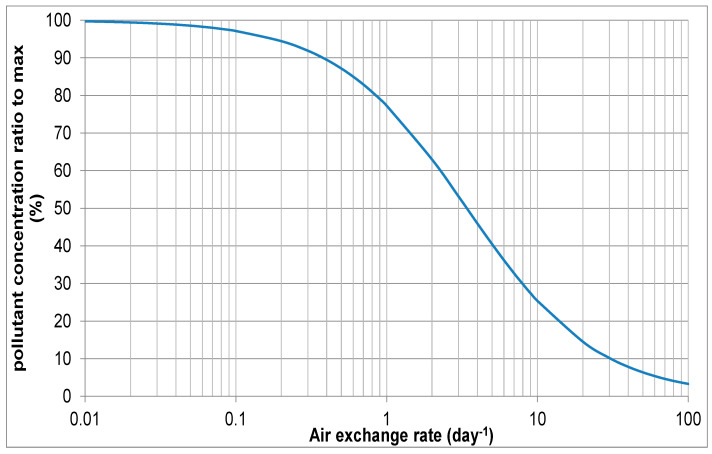
Effect of air exchange rate on formic acid concentration in the 0.08 AER showcase with a painted sign.

**Table 1 materials-16-07596-t001:** Techniques used or with potential to quantify corrosion on Oddy test coupons.

Technique	Used for	Notes	Reference
X-ray diffraction (XRD)	All	Grazing angle is required for Ag	[11]
Raman	All	Analytical area often small, large numbers of measurements required for representative results	[12]
Fourier transform Infra-red spectroscopy (FTIR)	Cu, Pb		[13,14]
Image capture and analysis	Cu, Ag		[8,15]
Image analysis of photographs	All		[16,17]
Mass Gain	Cu, Pb	Issues with different corrosion products providing different mass gains	[18]
Chemical stripping (mass loss)	All		[19]
Potentiodynamic stripping (mass loss)	Cu, Ag		[20]
Coulometric stripping (mass loss)	Cu, Ag		[21]
Scanning electron microscopy (SEM)	All		[22]
Colorimetry	All	Issues with early stages of Pb corrosion	[23,24,25]
X-ray photoelectron spectroscopy	Ag		[26,27]
Static Secondary ion mass spectrometry	Ag		[27]
Dynamic Secondary ion mass spectrometry	Ag		[27]
Oxygen depletion	Cu, Pb, (Fe)	Measurements of the test atmosphere and not the metal coupons	[28]

**Table 2 materials-16-07596-t002:** Main preventive conservation strategies for mitigating pollution damage.

Method	Description	Notes
Remove	Remove or, if testing, do not use materials determined to pose a risk	
Block	Cover the emissive surfaces with an impermeable material	Can be difficult to determine all emissive surfaces or can be inaccessible.
Dilute (passive)	Increased ventilation through holes	Can be difficult to predict and can vary with atmospheric conditions.
Dilute (active)	Forced ventilation with pump	Difficult to determine if pump running.
Sorb (passive)		Little information on amount required and difficult to calculate rate pollutant that sorbs onto the filter and hence the replacement period.
Sorb (active)	Forced filtered air with pump	Controlled flow rate, possible to predict replacement period. Some pumps have failure alarms.

**Table 3 materials-16-07596-t003:** Results from direct contact tests of materials that did not cause corrosion in Oddy tests.

Material	Number Tested	Additional Percentage as T or U for Materials That Were P for Copper, Silver and Lead in Normal Oddy Test		
		Copper	Silver	Lead
Powder-coated steel	10	10	10	10
Various fabrics from tests	50	16	22	18
Baumann Unisono	9	78	67	100
Baumann Usus	9	67	56	89
Baumann Velos	6	50	83	83

**Table 4 materials-16-07596-t004:** Results from emission tests of vinyls and calculations of when they exceed the low-risk threshold for lead and acetic acid at 50% RH. NA, not applicable, as the value already exceeded under-test conditions.

Sample	Smart Test Number	Acetic Acid Concentration (µg/m^3^)	Loading Needed to Exceed 400 µg/m^3^ Acetic Acid at Stated AER (per Day)				
			0.10	0.20	0.40	0.60	0.80
From Liverpool	Unknown	493	NA	NA	NA	NA	NA
White 8900	21/NCS001/070a1	130	23	32	50	69	87
Black 8900	21/NCS001/071a1	87	34	48	75	103	131
White 9800	21/NCS001/074a1	230	13	18	28	39	49
Graphite 8900	21/NCS001/072a1	465	NA	NA	NA	NA	NA
Black gloss 9800	21/NCS001/080a1	187	16	22	35	48	60

**Table 5 materials-16-07596-t005:** Instances of objects causing corrosion on metal objects.

Metal Corroding/Source	Corrosion Products	AER	Concentration (µg/m^3^)	Number Instances Observed (Reported)
Cu/ arch wood, ceramics, iron	Copper sulfides/sulfates	2.54	ND	32 (hundreds)
		1.12	ND	4
Ag/arch wood, ceramics	Silver sulfide	0.89	ND	3 (12)
Lead, wood	Basic lead carbonate			1 (approx 150)
		0.45	1234 (acetic)	2
		0.65	824 (acetic)	4
		0.89	1039 (acetic)	2
		0.91	864 (acetic)	1
		1.12	924 (acetic)	1
		1.34	724 (acetic)	1
		1.76	783 (acetic)	3
		3.21	643 (acetic)	1
Lead, paint	Lead formate	0.12	840 (formic)	
Silver/leather	Silver sulfide	1.78	ND	5
Copper alloy/oil paint	Copper formate	0.54	1612 (formic)	2
Copper alloy/wood	Copper acetate	0.12	1724 (acetic)	1
Copper/ceramic	Copper hydroxide	0.87	653 ammonia	2 (2)

**Table 6 materials-16-07596-t006:** Metal corrosion observed where the pollutant originated in the same object.

Metal Corroding/Source in Object—Object Type	Corrosion Products	AER	Concentration (µg/m^3^)	Number Instances Observed (Reported)
Lead/wood—Topcali figure Japanese lacquer boxes	Basic lead carbonate “	1.23 0.87 1.47 2.21 3.45	1047 Acetic 2564 Acetic 1254 1034 824	1 1 (64) 2 1 4
Silver/wool—battens Osborne House, medals Wellington Arch, sword, rode, cap	Silver sulfide “ “	26.73 0.40, 1.62 0.8, 1.2, 2.5	0.043 hydrogen sulfide <0.10 hydrogen sulfide <0.10 hydrogen sulfide	23 (approx 140) 3, 1 1, 2, 1
Iron/plastic, box of stencils	Iron nitrate	6.72	431 Nitrate-nitric	4
Copper/plastic, necklace	Copper nitrate	1.34	123 nitrate	1
Silver/felt Daguerretoypes/packets	Silver sulfide, Sodium formate on glass	0.4–1.2 Oxygen AERs		9
Aluminium/plywood drawers	Aluminium formate	0.56	1200 formic 800 acetic	7 (18)

**Table 7 materials-16-07596-t007:** Sulphur gas concentrations determined inside Tedlar bags enclosing archaeological artefacts.

Sample	Gas Concentration (µg/m^3^)		
	Hydrogen Sulfide	Carbonyl Sulfide	Carbon Disulfide
Empty bag	0.043	0.425	0.014
Ceramic 25 °C	0.184	2.481	0.351
Ceramic 35 °C	0.653	6.482	1.347
Iron 25 °C	0.248	1.234	0.012
Iron 35 °C	0.813	8.439	0.015

**Table 8 materials-16-07596-t008:** Results of mitigation support tool for selected instances from Table 5.

Measure Possible, Percentage Likely to Succeed						
	AER	Conc	Dilute—Passive	Dilute—Active	Sorb	Filter
Lead, wood	0.45	1234	YES, 50	YES, 100	YES, 20	YES, 100
Lead, wood	0.65	824	NO	YES, 100	YES, 90	YES, 100
Lead, wood	0.89	1039	YES, 80	YES, 100	YES, 70	YES, 100
Lead, wood	0.91	384	YES, 20	YES, 100	YES, 20	YES, 100
Lead, wood	1.12	924	NO	YES, 100	NO	YES, 100
Lead, wood	1.34	724	NO	YES, 100	YES, 10	YES, 100
Lead, wood	1.76	783	NO	YES, 70	NO	YES, 50
Lead, wood	3.21	643	NO	YES, 30	YES, 70	YES, 30
Lead, paint	0.12	840	YES, 50	YES, 100	NO	YES, 100

**Table 9 materials-16-07596-t009:** In-depth results from mitigation decision support tool and additional carbon footprint information from the method. Described by Thickett.

Action	Possible	Success	Initial Cost	Ongoing Cost	Carbon Footprint	Carbon Footprint (kg CO_2_ eq)
Reduce RH passive (silica gel)	YES	100	Low–med	v.low	Low–med	48
Reduce RH active (Munters MG50 dehumidifier)	YES	100	Med	Low	Med	113
Reduce pollution avoid	NO	N/A				
Reduce pollution block	NO	N/A				
Reduce pollution dilute passive	NO	N/A				
Reduce pollution dilute active	YES	75	V.low–low	Low	Med	42
Reduce pollution sorb	YES	20	Low–med	Low		12
Reduce pollution filter	YES	100	Low–high	Low	Med	106

## Data Availability

Data are available from the corresponding author.

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
