# Peer review of "Pollution and Heritage Metals—Effectiveness of Oddy Testing and Mitigation"

_materials, 2023, doi:10.3390/ma16247596_

Round 1
Reviewer 1 Report
Comments and Suggestions for Authors
The article deals with the Oddy test in such detail that "Oddy test" should be included in the title.
In Introduction (lines 24-26), could it maybe be emphasized that the corrosive atmospheres presented by showcases is due to offgassing from materials? As it is described in it is only implied in the text.
Introduction, line 90: In the sentence "Cano et al have suggested the use of coulometric stripping as per ISO 11844 [29]", the reference 29 is for the ISO standard, but should logically be for Cano et al, which, by the way must be Cano and Diaz (ref. 21)?
Following this, there is some confusion regarding lead and ISO 11844 (lines 93-96): While lead is not covered with regard to electrolytic cathodic reduction, chemical stripping is actually covered in the 2020 version with regard to mass loss measurement by chemical pickling of lead in an ammonium acetate solution. This information is btw also missing in the Cano and Diaz paper, so maybe the inaccurate statement origins from citing that paper?
In the Abstract is is stated that "..a decision support model has been developed" (line 20). However, was the decision support model tool developed for the study behind this current paper? It seems from the text that it origins from an earlier research project (Memory) and now applied to the current project's results? The Memory project is not mentioned as a funding source (line 551). Please clearify.
The link is reference 45 (English Heritage) doesn't work (Error 404 File not found)
Author Response
Thankyou for your comments, have addressed as below
Title amended as suggested
intro p24-29 text added re materials offgassing
p90, sorry a typo, have added 21, thought good to keep 29 here aswell
information added on chemical stripping of lead
more text added to clarify the decision support model, developed in MEMORI porject (now acknowledged), but expanded significantly afterwards
reference 45 does now work, (annoyingly that website only allows referencing to certain level, hence text in capitals as direction)
Reviewer 2 Report
Comments and Suggestions for Authors
This paper by Thickett and co-worker thoroughly reviews and discusses the complexity of the effect of pollutant gases to heritage metals. Through varous tests and analysis of samples, the authors investigates many aspects of this topic including sample preparation, quantification, analysis methods, etc.
One big problem of this paper is its structuring and phrasing. The authors are trying to including too much information in one paper, so the Introduction and the Conclusion parts read just like listing isolated items one by one (as can be seen from many short and weakly correlated paragraphs), instead of writing a complete and correlated article. Also, some of the writings are not scientific enough; some examples are below.
1. The authors used a lot of "probably," which should be minimized; if the authors are not confident about a statement, then it's better not to be included.
2. Many abbreviations did not come with a explanation, such as MEMORI project and SMART tests.
3. Occurances of "Powerpoint" and "spreadsheet" seems too trivial for a scientific paper. If required, the key parts can be replaced by formula/equation or decision tree/flow chart instead.
4. Some editoral updates such as inconsistent fonts in tables. Also some typos, such as Joel 840, should be Jeol 840 instead.
5. All raw data such as spectra and images should be included in the supporting materials.
In general, this is an interesting paper with comprehensiveness that is suitable for this special issue Corrosion Studies on Metallic Cultural Heritage. However, the structuring and the writing of this article needs to be improved so that it can meet the standard of Materials journal and cater to the general scientific community.
Please refer the the comments in the first section.
Author Response
Thankyou for the review
I have added more text and discussion to tie the introduction (including aims) and conclusion together more and explain how this series of investigations fits together. I have also altered eth structrure with another sub heading for clarity.
1, the use of probably has been significantly reduced, 3 instances remain, risk tolerated is probably institution dependant, there is no way to test this beyond and extensive survey . It is likely training could probably reduce variation in results and similar in conclusion, impossible to know for sure without testing, but the fact two groups have better grouping, indicates it may work.
2, Have added trext for the acronym MEMORI,
SMART test does not appear to be described anywhere, certainly not on their website
all other abbreviations have been expanded and bracketted if used later in text.
3, I have reworded powerpoint and added text why we used these particular tools. Cannot find a ready replacement for spreadsheet. Unfortunately each spreadsheet contains 3-15 formula and the decision tree incorporated in the powerpoint runs to 36 pages. With sufficient explanations, these would be would be an extremely large additional information, all can viewed readilly by downloading and the formula/decision tree are visible.
4, corrected and text reread
5 There are no supporting images. There are over 800 FTIR spectra, 16 per coupon for full surface coverage. All identified as lead carbonate or cuprite and copper formate. I do not see any particular benefit in providing these. As stated in the data availability statement, I am happy to send the spectra to anyone that requests, as I can do this via bulk file transfer, without having to combine into a single enormous PDF
Reviewer 3 Report
Comments and Suggestions for Authors
Title
Pollution and heritage metals
David Thickett, Tayba Batool
Abstract
Lines 8-20
Pleaase structure as:
Introduction-Aims
Method
Results and interpretation
Introduction lines 20-183
Please develop the subchapter. It is too short
Please develop the literature review section and update to 2023
Please analyses critically the findings of the articles and the limitations.
Please indicate also at least tree similar article to your research published recently (last 5 years).
Scope of your research is not very clear expressed
Tables 1 and 2 please add the datasource
Methodology
Some recent citations please to be included.
Limitations of the study should be inserted
Please add the datasources and indicate what repreasent each component for the equations
References list: enlarge and internationalize please the references list.Update it to 2023.
We consider that can be useful for the paper also the following reference:
DOI
10.35530/IT.072.01.1824
DOI
10.25083/rbl/25.2/1362.1368
The paper reflects the title and can be useful for the journal readers as well as for practitioners. But the references list must be improved: updated to 2023, internationalized as well.
Is an actual topic even it is not very original; the subject is economically and recycling technology very important. Several materials present in artefacts are known to potentially cause corrosion when enclosed with metals in other artefacts. Have been investigated some situations with pollution and RH and some corrosion rate measurements. Ways to isolate artefacts or mitigate are explored and a decision support model has been developed
The paper should be much better documented (very few published articles in references list), very few … and also very few updated to 2023.
References Should be revised.
The paper is well written. The quality of English translation is very good.
The text is well structured, clear and easy to read from the specialists in the field but as well as from the persons from public.
thanks a lot,
November 2023
Author Response
thankyou for your useful comments
have renamed the sections and added aims. I have kept conclusion as I feel this is useful to the reader.
I may not understand your point re intrsoduction, its 160 lines long, so can't see that expansion is required/desireable.
I have indicated were info in tables 1 and 2 originated.
The paper has 50 references and 14 from the last 5 years. I have checking BCIN, AATA databases and article lists for JAIC, Studies in Conservation, Heritage, Heritage Science, Journal of Cultural Heritage, along with IIC, AIC and ICOM-CC proceedings and failed to find any further recent relevant publications. The two DOIs you suggest do not appear to be related to this topic.
The limitations of the methods are discussed in some detail in the text and I am not clear what extra you are suggesting as additions.